# Bridging the Gap between Human Motion and Action Semantics via Kinematic Phrases

## Abstract

The goal of motion understanding is to establish a reliable mapping between motion and action semantics, while it is a challenging many-to-many problem. An abstract action semantic (i.e., *walk forwards*) could be conveyed by perceptually diverse motions (walk with arms up or swinging), while a motion could carry different semantics w.r.t. its context and intention. This makes an elegant mapping between them difficult. Previous attempts adopted direct-mapping paradigms with limited reliability. Also, current automatic metrics fail to provide reliable assessments of the consistency between motions and action semantics. We identify the source of these problems as the **significant gap** between the two modalities. To alleviate this gap, we propose Kinematic Phrases (KP) that take the objective kinematic facts of human motion with **proper abstraction**, **interpretability**, and **generality** characteristics. Based on KP as a mediator, we can unify a motion knowledge base and build a motion understanding system. Meanwhile, KP can be **automatically** converted from motions and to text descriptions with no subjective bias, inspiring Kinematic Prompt Generation (KPG) as a novel automatic motion generation benchmark. In extensive experiments, our approach shows superiority over other methods. *Our code and data would be made publicly available*.

## 1 Introduction

Human motion understanding has a wide range of applications, including autonomous driving (Paden et al., 2016), robotics (Koppula & Saxena, 2013), and automatic animation (Van Welbergen et al., 2010), making it increasingly attractive. The core of human motion understanding is to establish a mapping between the motion space and the action semantics space. The motion space indicates a space of sequential 3D human representations, e.g., 3D pose or SMPL (Loper et al., 2015)/SMPL-X (Pavlakos et al., 2019) parameter sequence, while the action semantic space can be represented as action categories or sentences described by natural language.

Recently, a growing focus has been on generative mapping from semantics to motion, including action category-based generation (Petrovich et al., 2021) and text-based generation (Petrovich et al., 2022; Guo et al., 2022a; Lucas et al., 2022; Zhang et al., 2022; Tevet et al., 2022b; Chen et al., 2023; Zhang et al., 2023a). Most of them typically build a mapping that links motion and semantics either directly or via motion latent, with understated concerns for intermediate motion-semantic structures. However, these models suffer from inferior reliability. They cannot guarantee they generated correct samples without human filtering. Additionally, the existing evaluation of motion generation is problematic. Widely adopted FID and R-Precision rely on the latent space from a black-box pre-trained model, which might fail to out-of-distribution (OOD) and over-fitting cases. There is a long-standing need for an evaluation method that can cheaply and reliably assess whether a generated motion is consistent with particular action semantics. We identify the essence of these as the significant gap between raw human motion and action semantics, which makes direct mapping hard to learn.

As in Fig. 1, an action semantics can correspond to diverse motions. For instance, a person could *walk* in countless ways with diverse motions, either with arms up or swinging, while action semantics tend to abstract these away from a walking motion. Additionally, they are robust against small perturbations, while motion is more specific and complex, with representations changing vastly when perturbed or mis-captured. Moreover, a motion sequence could have diverse semantics w.r.t. contexts. Modeling this many-to-many mapping between motion and semantics is challenging.

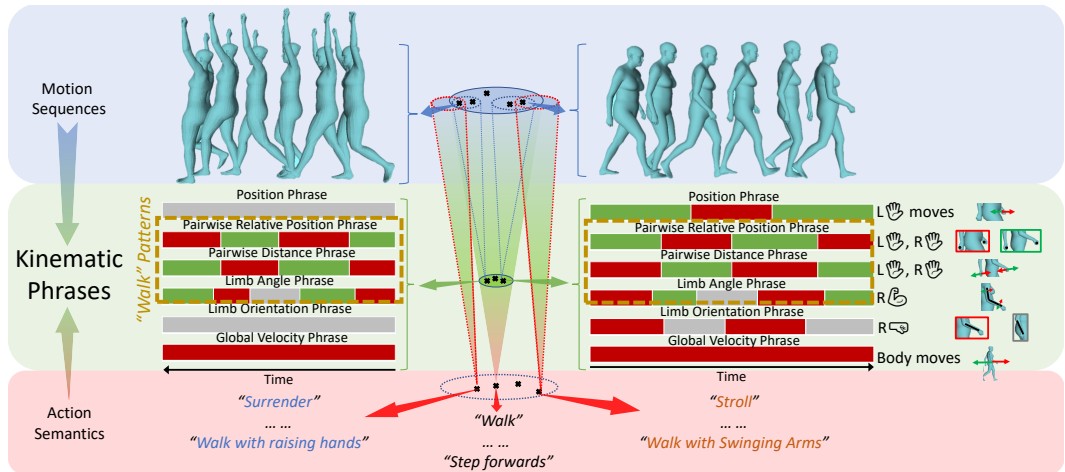

Figure 1: The huge gap between motion and action semantics results in the *many-to-many* problem. We propose Kinematic Phrases (KP) as an intermediate to bridge the gap. KPs objectively capture human kinematic cues. It properly abstracts diverse motions with interpretability. As shown, the Phrases in the yellow box could capture key patterns of *walk* for diverse motions.

To bridge this gap between motion and action semantics, we propose Kinematic Phrases (KP), an interpretable intermediate representation. KP focuses on the objective kinematic facts, which are usually omitted by general action semantics, like `left-hand moving forwards then backward`. KP is designed as qualitative categorical representations of these facts. For objectivity and actuality, KP captures **sign changes** with minimal pre-defined standards. Inspired by previous studies on kinematic human motion representation (von Laban & Lange, 1975; Bartlett, 1997), KP is proposed as six types shown in Fig. 1, covering **joint positions**, **joint pair positions** and **distances**, **limb angles** and **directions**, and **global velocity**. Note that, although KP can be described by natural language, a major difference is that KP is strictly dedicated to objective kinematic facts instead of coarse actions such as *surrender* or fine-grained actions like *raise both hands*.

We highlight three advantages of KP. First, KP offers **proper abstraction**, which disentangles motion perturbations and semantics changes, easing the learning process. Even though the motion differs significantly, KP manages to capture *walk* patterns easily. Second, KP is **interpretable**, as it can be viewed as instructions on executing the action, making it easily understandable to humans. Finally, KP is **general**, as it can be automatically extracted from different modalities of human motion, including skeleton and SMPL parameters. The conversion from KP to text is also effortless.

With KP as an intermediate representation, we first construct a unified large-scale motion knowledge base. Then, to fully exploit KP and the knowledge base, we build a motion understanding system with KP mediation. In detail, we learn a motion-KP joint latent space in a self-supervised manner and then adopt it for multiple motion understanding applications, including motion interpolation, modification, and generation. Moreover, leveraging the interpretability of KP, we propose a benchmark called Kinematic Prompts Generation (KPG), which generates motion from text prompts converted from KPs. Thanks to the consistency and convenience of the KP-to-text conversion, KPG enables reliable and efficient motion generation evaluation.

Our contributions are: (1) We propose KP as an intermediate representation to bridge the gap between motion and action semantics. (2) We build a novel motion understanding system using KP and the aggregated large-scale knowledge base. (3) We propose KPG as a benchmark for reliable and efficient motion generation evaluation. Promising results are achieved on motion interpolation and generation tasks. Moreover, extensive user studies are conducted, verifying the efficacy of our methods, also the consistency between KPG evaluation and human perception.

## 2 RELATED WORKS

**Motion Representation**. An intuitive motion representation is a sequence of static pose representations, like joint locations and limb rotations. Efforts are paid to address the discontinuity of

rotation for deep-learning methods (Zhou et al., 2019; Brégier, 2021). Recent works on parametric body models (Loper et al., 2015; Pavlakos et al., 2019) enable a more realistic body representation. Meanwhile, Pons-Moll et al. (2014) proposed Posebits, representing pose with boolean geometric part relationships. Delmas et al. (2022; 2023) translates Posebits into text descriptions. These abstract representations are flexible and insensitive to little perturbations, but their static nature ignores motion dynamics. Tang et al. (2022) acquire similar fine-grained descriptions from human annotation, while Xiang et al. (2022); Athanasiou et al. (2023) adopted large-scale language models. However, few recognize their potential in bridging the low-level motion and the high-level action semantics. Phase functions (Holden et al., 2020), Labanotations (von Laban & Lange, 1975), and learned Motion Words (Aristidou et al., 2018) were also explored, though limited to specific actions like locomotion and dancing.

**Motion Generation** can be conditioned by its prefix/suffix (Hernandez et al., 2019; Athanasiou et al., 2022; Guo et al., 2023), action categories (Petrovich et al., 2021; Guo et al., 2020; Xu et al., 2023), or audio (Li et al., 2021a;b). Text-based motion generation has developed rapidly with the proposal of text-motion datasets Punnakkal et al. (2021); Guo et al. (2022a). Petrovich et al. (2022); Guo et al. (2022a); Qian et al. (2023) used VAEs, while Tevet et al. (2022a); Hong et al. (2022); Lin et al. (2023b) extended the CLIP (Radford et al., 2021) space to motion. Recently, attention has been paid to diffusion models (Zhang et al., 2022; Tevet et al., 2022b; Dabral et al., 2023; Wang et al., 2023). Azadi et al. (2023) adopted a U-Net structure. Zhang et al. (2023b); Petrovich et al. (2023) explored retrieval-based methods. Karunratanakul et al. (2023) aimed at controllable generation, while Yuan et al. (2023) introduced physical constraints. However, most approaches still suffer from the gap between motion and action semantics. Lucas et al. (2022); Guo et al. (2022b); Zhang et al. (2023a); Chen et al. (2023); Zhou & Wang (2023); Zhong et al. (2023); Kong et al. (2023) adopted (VQ-)VAE-compressed motion representation as mediation, while in the current data-limited situation, we identify that this single-modality compression might be sub-optimal. Instead, KP could alleviate this by introducing explicit semantic-geometric correlation.

## 3 KINEMATIC PHRASE BASE

### 3.1 KINEMATIC PHRASES

Kinematic Phrases abstract motion into objective kinematic facts like `left-hand moves up` qualitatively. We take inspiration from previous kinematic motion representations (von Laban & Lange, 1975) and qualitative static pose representations (Delmas et al., 2022; Pons-Moll et al., 2014), proposing six types of KP to comprehensively represent motion from different kinematic hierarchies: For **joint movements**, there are 36 Position Phrases (PPs). For **joint pair movements**, there are 242 Pairwise Relative Position Phrases (PRPPs) and 81 Pairwise Distance Phrases (PDPs). For **limb movements**, there are 8 Limb Angle Phrases (LAPs) and 33 Limb Orientation Phrases (LOPs). For **whole-body movements**, there are 3 Global Velocity Phrases (GVPs). KP extraction is based on a skeleton sequence $X = \{x_i | x_i \in \mathcal{R}^{n_k \times 3}\}_{i=1}^t$, where $n_k$ is the number of joints ($n_k = 17$ here), $x_i$ is the joint coordinates at $i$-th frame, and $t$ is the sequence length. Note that $x_i^0$ indicates the pelvis/root joint. For each Phrase, a scalar indicator sequence is calculated from the skeleton sequence. Phrases are extracted as per-frame categorical representations w.r.t. indicator signs. Unlike previous efforts (Pons-Moll et al., 2014; Delmas et al., 2022), we limit the criteria of KP as the indicator signs to minimize the need for human-defined standards (e.g., numerical criteria on the closeness of two joints) for objectivity and actuality. Fig. 2 illustrated the extraction procedure.

**Reference Vectors** are first constructed, indicating right, upward, and forward directions from a human *cognitive view*. We aim at the *egocentric* reference frames that human tends to use when performing actions. The negative direction of gravity is adopted as upward vector $r^u$, the vector from left hip to right hip is adopted as right vector $r^r$, and the forward vector is calculated as $r^f = r^u \times r^r$. These vectors of each frame are denoted as $R^{\cdot} = \{r_i^{\cdot}\}_{i=1}^t$.

**Position Phrase (PP)** focuses on the movement direction of joint $x^j$ w.r.t. reference vector $R^{\cdot}$. The indicator for PP at $i$-th frame is calculated as

$$s_i^{(j,\cdot)} = \langle (x_i^j - x_i^0), r_i^{\cdot} \rangle - \langle (x_{i-1}^j - x_{i-1}^0), r_{i-1}^{\cdot} \rangle. \tag{1}$$

The sign of $s_i^{(j,\cdot)}$ categorizes PP into moving `along`/`against` $R^{\cdot}$, or `relatively static` along $R^{\cdot}$ for indicators with small amplitudes. After filtering, 36 different PPs are extracted.

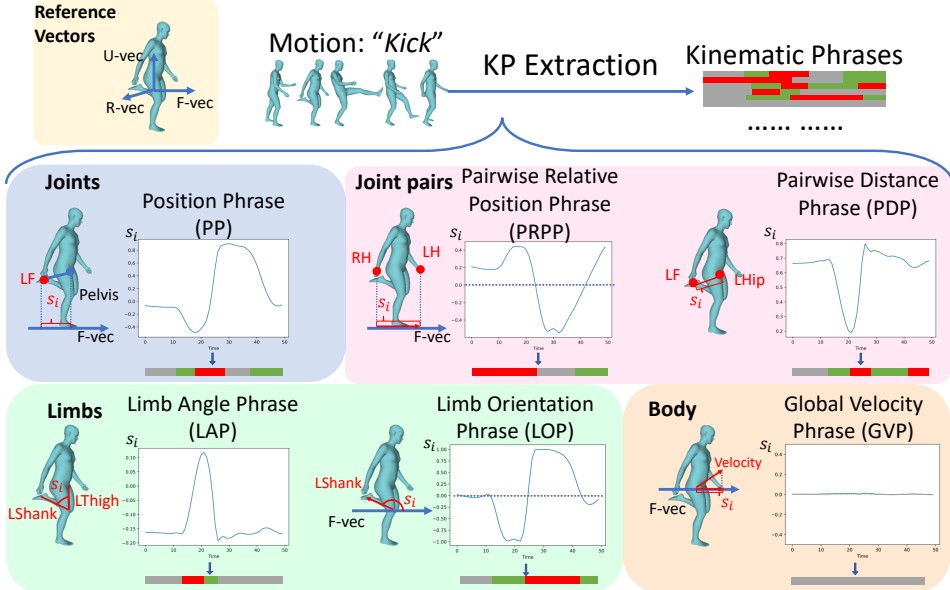

Figure 2: Six types of KP from four kinematic hierarchies are extracted from a motion sequence. A scalar indicator $s_i$ is calculated per Phrase *per frame*. Its sign categorizes the corresponding Phrase.

**Pairwise Relative Position Phrase (PRPP)** describes the relative position between a pair of joints $(x^j, x^k)$ w.r.t. reference vector $R^{\cdot}$. PRPP indicator at $i$-th frame is $s_i^{(j,k,\cdot)} = \langle (x_i^j - x_i^k), r_i^{\cdot} \rangle$. For (L-Hand, R-Hand) and forward vector $R^f$, PRPP could be L-Hand behind/in front of R-Hand according to the sign of $s_i^{(j,k,\cdot)}$. After filtering, 242 PRPPs are extracted.

**Pairwise Distance Phrase (PDP)** describes how the L2 distance between a pair of joints $(x^j, x^k)$ changes. The indicator for PDP is calculated as

$$s_i^{(j,k)} = \|x_i^j - x_i^k\|_2 - \|x_{i-1}^j - x_{i-1}^k\|_2. \tag{2}$$

The sign of $s_i^{(j,k)}$ categorizes PDP into moving closer/away, or relatively static. After dropping joint pairs in the skeleton topology, such as the hand and elbow, 81 PDPs are extracted.

**Limb Angle Phrase (LAP)** targets at the change of bend angle between two connected limbs $(x^j, x^k)$ and $(x^j, x^l)$. The indicator for LAP is calculated as

$$s_i^{(j,k,l)} = arccos(\langle x_i^k - x_i^j, x_i^l - x_i^j \rangle) - arccos(\langle x_{i-1}^k - x_{i-1}^j, x_{i-1}^l - x_{i-1}^j \rangle). \tag{3}$$

LAP describes the limb chain $(x^j, x^k)$-$(x^j, x^l)$ as bending or unbending. 8 LAPs are extracted.

**Limb Orientation Phrase (LOP)** describes the orientation of the limb $(x^j, x^k)$ w.r.t. $R^{\cdot}$, note that $x^k$ is the distal limb. The scalar indicator for LOP is calculated as $s_i^{(j,k,\cdot)} = \langle x_i^k - x_i^j, r_i^{\cdot} \rangle$. The sign of $s_i^{(j,k,\cdot)}$ categorizes the LOP into limb $(x^j, x^k)$ pointing along/against $R^{\cdot}$, or a placeholder category for those with little magnitude. 33 LOPs are extracted.

**Global Velocity Phrase (GVP)** describes the direction of global velocity with respect to $R^{\cdot}$. The indicator is calculated as $s_i^{\cdot} = \langle x_{i+1}^0 - x_i^0, r_i^{\cdot} \rangle$. The three categories are moving along/against $R^{\cdot}$, or static along $R^{\cdot}$ according to the sign of $s_i^{\cdot}$.

These result in 403 Phrases in total, covering motion diversity and distribution from various levels. While we clarify that these Phrases do not rule out the possibility of other possible useful potentials.

## 3.2 CONSTRUCTING KINEMATIC PHRASE BASE

KP enables us to unify motion data with different formats to construct a large-scale knowledge base containing motion, text, and KP. Motion sequences of different representations are collected, including 3D skeleton sequences and SMPL (Loper et al., 2015)/SMPL-X (Pavlakos et al., 2019) parameter sequences. The sequences are first re-sampled to 30Hz and rotated so that the negative

direction of the z-axis is the gravity direction. Then, the sequences are converted into 3D skeleton sequences for KP extraction as in Sec. 3.1. Text annotations attached to the sequences are directly saved. For sequences with action category annotation, the category name is saved. For those with neither text nor action category, the text information is set from its attached additional information, like objects for SAMP (Hassan et al., 2021). Finally, we collect 87k motion sequences from 11 datasets. Detailed statistics are shown in Tab. 1. More details are included in the appendix.

# 4 MOTION UNDERSTANDING VIA KP

By motion understanding, we mean both low-level understanding like interpolation and modification, and high-level understanding like generative mapping from text to motion. To achieve this, we first learn a motion-KP joint space with less ambiguity and more interpretability. Then, with this space, we introduce its application to both low-level and high-level motion-semantics understanding.

| Dataset | Mot. Rep. | #Seqs | #Actions | Text |
|---|---|---|---|---|
| AMASS (Mahmood et al., 2019) | SMPL-X | 26k | 260 | ✓ |
| GRAB (Taheri et al., 2020) | SMPL-X | 1k | 4 | ✓ |
| SAMP (Hassan et al., 2021) | SMPL-X | 0.2k | N/A | ✓* |
| Fit3D (Fieraru et al., 2021) | SMPL-X | 0.4k | 29 | ✓ |
| CHI3D (Fieraru et al., 2020) | SMPL-X | 0.4k | 8 | ✓ |
| UESTC (Ji et al., 2018) | SMPL | 26k | 40 | ✓ |
| AIST++ (Li et al., 2021a) | SMPL | 1k | N/A | ✓* |
| BEHAVE (Bhatnagar et al., 2022) | SMPL | 0.3k | N/A | ✓* |
| HuMMan (Cai et al., 2022) | SMPL | 0.3k | 339 | ✓ |
| GTAHuman (Cai et al., 2021) | SMPL | 20k | N/A | x |
| Motion-X(Lin et al., 2023a) | SMPL-X | 65k | N/A | ✓ |
| **Sum** | - | **140k** | **680+** | - |

Table 1: Statistics of Kinematic Phrase Base. *Mot. Rep.* indicates motion representation. "✓*" means texts are generated from the attached additional information instead of human annotation.

## 4.1 PRELIMINARIES

We first introduce the representation for motion and KP. **Motion** is represented as a human pose sequence with $n$ frames as $M = \{m_i\}_{i=1}^n$. In detail, SMPL (Loper et al., 2015) pose parameters are transformed from axis-angle format to the 6D continuous representation (Zhou et al., 2019), then concatenated with the velocity of the root joint, resulting in a 147-dimensional representation per frame. **KP** is represented by signs of the indicators.

## 4.2 JOINT SPACE LEARNING

**Model Structure.** An overview of our model is illustrated in Fig. 3. **Motion VAE** is a transformer-based VAE adapted from Petrovich et al. (2021). The encoder $\mathcal{E}_m$ takes motion $M$ and two distribution tokens $m_\mu, m_\sigma$ as input, and the outputs corresponding to the distribution tokens are taken as the $\mu_m$ and $\sigma_m$ of the Gaussian distribution. Then, the transformer decoder $\mathcal{D}_m$ takes $z_m \sim \mathcal{G}(\mu_m, \sigma_m)$ as $K, V$, and a sinusoidal positional encoding of the expected duration as $Q$. The output is fed into a linear layer to obtain the reconstructed motion sequence $\hat{M}$. **KP VAE** with encoder $\mathcal{E}_p$ and decoder $\mathcal{D}_p$ resembles Motion VAE. The sign of $\mathcal{D}_p$ output is adopted as the predicted KP $\hat{C}$. Notice that the decoders $\mathcal{D}_m, \mathcal{D}_p$ could take arbitrary combinations of $z_m, z_p$ as input, outputting $\hat{M}., \hat{C}.$.

**Self-supervised Training.** With the VAEs, we propose a self-supervised training strategy to learn motion-KP joint space. As a coherent representation, the overall representation should not change drastically with a small portion of KP unknown. Even more, the missing Phrases should be recovered from existing Phrases. In this view, we randomly corrupt samples during training by setting a small portion of KP as 0. The training is thus executed in a self-supervised manner. This helps mine the correlation among different Phrases while also effectively increasing the robustness of the joint space. Similar to TEMOS (Petrovich et al., 2022), four losses are adopted: reconstruction loss, KL divergence loss, distribution alignment loss, and embedding alignment loss.

## 4.3 KP-MEDIATED MOTION UNDERSTANDING

With the joint space, we can perform both low-level and high-level motion understanding with KP mediation. We introduce three applications to show the capability of KP, as shown in Fig. 3.

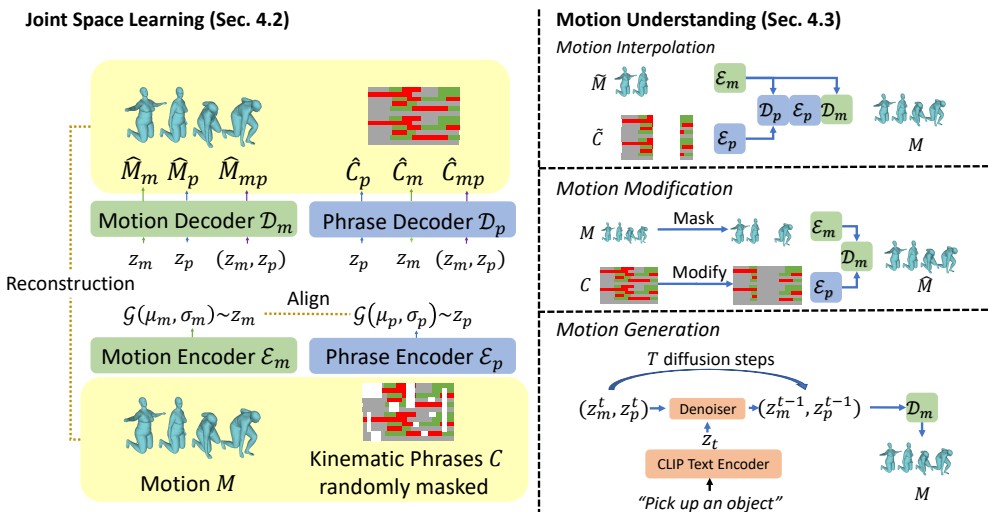

Figure 3: We train motion-KP joint latent space in a self-supervised training manner. KP is randomly masked during training. Reconstruction and alignment losses are adopted. The joint space could be applied for multiple tasks, including motion interpolation, modification, and generation.

**KP-mediated Motion Interpolation** Given a corrupted motion sequence $\tilde{M}$, we extract its corresponding KP sequence $\tilde{C}$, then feed them to encoders $\mathcal{E}_m, \mathcal{E}_p$ and decoder $\mathcal{D}_p$, resulting in the estimated KP sequence $\hat{C}$. $\hat{C}$ and $\tilde{M}$ are fed into $\mathcal{E}_m, \mathcal{E}_p$ and $\mathcal{D}_m$, resulting in interpolated $\hat{M}$.

**Motion Modification** Motion modification functions similarly. Motion $M$ is first extracted into KP sequence $C$. Modifications could be made on $C$ resulting in $\tilde{C}$. Modified motion frames are then masked, getting $\tilde{M}$. $\tilde{M}, \tilde{C}$ are fed into $\mathcal{E}_m, \mathcal{E}_p$ and $\mathcal{D}_m$, getting the interpolated $\hat{M}$.

**KP-mediated Motion Generation.** Given text $t$, to generate a motion sequence from it, we first encode it into latent $z_t$ with CLIP text encoder $\mathcal{E}_t$. Direct mapping could be achieved by training the motion decoder $\mathcal{D}_m$ for $\hat{M} = \mathcal{D}_m(z_t)$. We show that the direct mapping could be impressively improved with our joint space in Sec. 6.4. With KP, we could perform a novel KP-mediated motion generation. We adopt a vanilla latent diffusion paradigm for KP-mediated text-to-motion tasks. An extra denoiser is trained to denoise a random noise $z_p^T$ to KP latent $z_p = z_p^0$ with $T$ diffusion steps. We then decode KP sequence $\hat{C}$ from $z_p$ with $\mathcal{D}_p$. Then, $\hat{C}$ is encoded by $\mathcal{E}_p$, getting distribution $\mathcal{G}(\mu_p, \sigma_p)$. $z_p$ is sampled and sent to $\mathcal{D}_m$ to generate a motion sequence. Experiments show that KP could be a promising stepping stone to mitigate the huge gap from action semantics to motion.

## 5 KINEMATIC PROMPT GENERATION

With the interpretability and objectivity of KP, we propose a new motion generation benchmark.

Before that, we first analyze current benchmarks. A crucial aspect of motion generation evaluation is motion-semantic consistency. The gold standard is user study. However, it is expensive and inefficient to scale. Early metrics like MPJPE (Mean Per Joint Position Error) and MAE (Mean Angle Error) mechanically calculate the error between the generated and GT samples. These metrics fail to reveal the real ability of generative models: What if the models memorize GT samples? Or what if the samples are diverse from GT but also true? FID (Frechet Inception Distance) is adopted to mitigate this issue. However, it provides a macro view of the quality of all generated samples without guarantees for individual samples. Guo et al. (2022a) proposed R-Precision, using a pre-trained text-motion matching model to examine whether the generated samples carry true semantics. They both rely on the latent space from a black-box pre-trained model, which is not credible. Besides, models might learn short paths to over-fit the pre-trained model. Moreover, since automatic mapping from motion to semantics across their huge gap is still an unsettled problem, adopting it to evaluate motion generation is not a decent choice. Moreover, most current motion generation evaluations are performed on datasets (Guo et al., 2022a; Plappert et al., 2016; Ji et al., 2018) with considerable complex everyday actions, further increasing the difficulty.

To this end, we propose a novel benchmark: Kinematic Prompts Generation (KPG). Instead of previous benchmarks focusing on everyday activities or sports, we take a step *back* in the complexity of the target action semantics. Based on KP, KPG focuses on evaluating *whether the models could generate motion sequences consistent with specific kinematic facts given text prompts*.

In detail, we convert KP into text prompts with templates as in Tab. 2, resulting in 840 text prompts. Given prompt $T_i \in T$ from Phrase $c_i$, the model generates motion $\hat{M}_i$, along with extracted KP $\hat{C}_i$. We calculate Accuracy as $Acc = \frac{1}{|T|} \sum_{T_i \in T} 1[c_i \in \hat{C}_i]$, where $1[\cdot] = 1$ if the expression in $[\cdot]$ is True, otherwise 0. Note that, for $c_i \in \hat{C}_i$, $c_i$ should keep for more than 5 consecutive frames to avoid trivial perturba-

| KP | Text prompt samples |
|---|---|
| PP | **Left hand** moves `forwards`. |
| PRPP | **Left hand** is `below` **head** then `above` **head**. |
| PDP | **Left hand** moves `away from` **head**. |
| LAP | **Left arm** `bends`. |
| LOP | **Left forearm** points `forwards` then `backward`. |
| GVP | The person moves `forwards`. |

Table 2: Text prompts converted from KP. **Joint/limb names**, *prepositions, verbs, and adverbials* could be replaced w.r.t. specific Phrases.

tions. Accuracy examines whether the Phrase corresponding to the given prompt appears in the KP sequence converted from generated motion. The calculation involves no black-box model thanks to KP, presenting a fully reliable evaluation pipeline. Also, with the effortless motion-to-KP conversion, the computation could be conducted automatically. More details are in the appendix.

## 6 EXPERIMENT

**Implementation Details.** HumanML3D (Guo et al., 2022a) test split is held out for evaluation, with the rest of KPB for training. During training, the motion sequences are canonicalized by eliminating the rotation along the z-axis in the first frame, and the same counter-rotation is applied to the following frames. Sequences are sampled to 15 FPS and randomly clipped into short clips with lengths between 30 frames and 150 frames. The batch size is set as 288, and an AdamW optimizer with a learning rate of 1e-4 is adopted. We randomly corrupt less than 20% of the Phrases for a sample. The Motion-KP joint space is trained for 6,000 epochs. While the text-to-motion latent diffusion model is trained for 3,000 epochs, with the joint space frozen. All experiments are conducted in 4 NVIDIA RTX 3090 GPUs. More details are provided in the appendix.

### 6.1 MOTION INTERPOLATION

Following Jiang et al. (2023), 50% frames are randomly masked for interpolation evaluation. FID and Diversity are also evaluated. We adopt MDM (Tevet et al., 2022b) as the baseline. In Tab. 3, our method provides better FID. While with additional KPB, the Diversity is increased.

### 6.2 MOTION GENERATION

**Settings**. We adopt the HumanML3D test set (Guo et al., 2022a) for conventional text-to-motion evaluation. The evaluation model from Guo et al. (2022a) is adopted to calculate R-Precision, FID, Diversity, and Multimodality. KPG is also adopted, with the proposed Accuracy. Also, Diversity is computed as a reference. We run the evaluation 20 times and report the average metric value. Details are given in the appendix.

**Results on conventional text to motion** are shown in Tab. 3. Our method is competitive without KPB. However, KPB brings a counter-intuitive performance drop. To evaluate this, we further conduct a user study to make human volunteers judge the motions instead of a proxy neural network.

Our user study is different from previous efforts in two aspects. First, instead of testing a small set of text prompts (less than 50 in previous works (Tevet et al., 2022b; Chen et al., 2023)), we randomly select 600 sentences from the HumanML3D test set. By scaling up, the result is convincing in reflecting the ability to generate motion for diverse text inputs. Second, neither asking the volunteers to give a general rating for each sample nor to choose between different samples, we ask them two questions: 1) Do the motion and the text match? and 2) Is the motion natural? For Q1, three choices are given as "No, Partially, Yes". For Q2, two choices are given as "Yes, No". In this way, we explicitly decouple the evaluation of text-to-motion into semantic consistency and naturalness, corresponding to R-Precision and FID. For each prompt, we generate one sample considering the

| Methods | Motion Interpolation | | Motion Generation | | | |
|---|---|---|---|---|---|---|
| | FID↓ | Diversity→ | R-P@1↑ | FID↓ | Diversity→ | Multimodality |
| GT | 0.002 | 9.503 | 0.511 | 0.002 | 9.503 | - |
| TEMOS (Petrovich et al., 2022) | - | - | 0.424 | 3.734 | 8.973 | 0.368 |
| T2M (Guo et al., 2022a)* | - | - | 0.455 | 1.067 | 9.188 | 2.090 |
| MDM (Tevet et al., 2022b)* | 2.698 | 8.42 | 0.320 | 0.544 | 9.559 | 2.799 |
| TM2T (Guo et al., 2022b)* | - | - | 0.424 | 1.501 | 8.589 | 2.424 |
| MLD (Chen et al., 2023)* | - | - | 0.481 | 0.473 | 9.724 | 2.413 |
| T2M-GPT (Zhang et al., 2023a)* | - | - | **0.492** | **0.141** | 9.722 | 1.831 |
| MotionGPT (Jiang et al., 2023)* | 0.214 | **9.560** | **0.492** | 0.232 | **9.528** | 2.008 |
| Ours* | **0.197** | 9.772 | 0.475 | 0.412 | 10.161 | 2.065 |
| Ours | 0.226 | 10.022 | 0.434 | 0.631 | 10.372 | 2.584 |

Table 3: Result Comparison of motion interpolation and generation on HumanML3D. R-P@1 is short for R-Precision@1. * indicates the model is trained on the HumanML3D train set only.

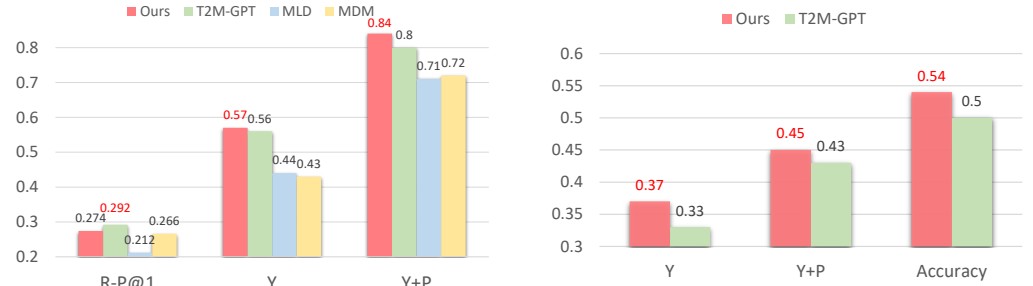

Figure 4: User study on HumanML3D, with "Y" for Yes and "P" for partially.

Figure 5: User study on KPG, with "Y" for Yes and "P" for partially.

annotation cost. We claim that the models should generate natural text-matching motion most of the time so that the one-sample setting would not hurt the fidelity of our user study. 36 volunteers are invited, each reviewing 200 sequences. Thus each sequence receives 3 user reviews. Also, we compute R-precision@1 of the generated sequences for reference. MDM (Tevet et al., 2022b), T2M-GPT (Zhang et al., 2023a), MLD (Chen et al., 2023), and our method are evaluated.

User study results are shown in Fig. 4. Though our method is not superior in R-Precision, we receive better user reviews, showcasing the efficacy of our KP-mediated generation strategy. Recent T2M-GPT and MLD present similar R-Precision, but only T2M-GPT manages to keep a good performance with user reviews. Moreover, the discrepancy between R-Precision and user reviews is revealed in both absolute value and trends. More results and analysis are given in the appendix.

**Results on KPG** are shown in Tab. 4. KPG is considered an easier task than conventional text-based motion generation since it is targeted at action semantics with much less complexity. However, previous methods are not performing as well as expected. Though we managed to deliver substantial improvements, the accuracy remains below 60%, which is far from satisfying. There is a considerable gap between existing methods and ideal motion generation models.

Furthermore, given the discrepancy between automatic metrics and user study as shown in Fig. 4, we conducted a similar user study with 100 randomly selected prompts from KPG involving T2M-GPT and our model. Fig. 5 demonstrates that KP-inferred Accuracy and user reviews share similar trends. We also calculate their consistency, showing KP and user study give the same reviews for **84%** of the samples. We believe KPG could thus be a first step towards reliable automatic motion generation evaluation. More analyses are given in the appendix.

### 6.3 VISUALIZATION

We first present a modification sample in Fig. 6. By modifying KP, we could edit arbitrary motion at a fine-grained level. Also, We compare generated samples of T2M-GPT and our methods in Fig. 7. Our method properly responds to text prompts with constraints on specific body parts. This could be attributed to KP mediation, which explicitly decomposes the action semantics into kinematics cues of body parts. Note that T2M-GPT might generate redundant motion for simple prompts, while our method provides more concise and precise results. More visualizations are in the appendix.

| Methods | Acc.%↑ | Diversity |
|---|---|---|
| HMDM (Tevet et al., 2022b) | 44.40 | 5.725 |
| MLD (Chen et al., 2023) | 44.76 | 5.901 |
| T2M-GPT (Zhang et al., 2023a) | 47.86 | 6.593 |
| Ours | **52.14** | 6.017 |

Table 4: Results on Kinematic Prompt Generation.

| Methods | Acc.%↑ | Diversity |
|---|---|---|
| Ours | **52.14** | 6.017 |
| w/o KP mediation | 50.43 | 5.616 |
| Direct mapping | 42.28 | 5.379 |
| w/o Joint KP | 51.03 | 5.765 |
| w/o Joint Pair KP | 48.24 | 5.596 |
| w/o Limb KP | 51.92 | 5.804 |
| w/o Body KP | 52.04 | 5.903 |

Table 5: Ablation results on KPG.

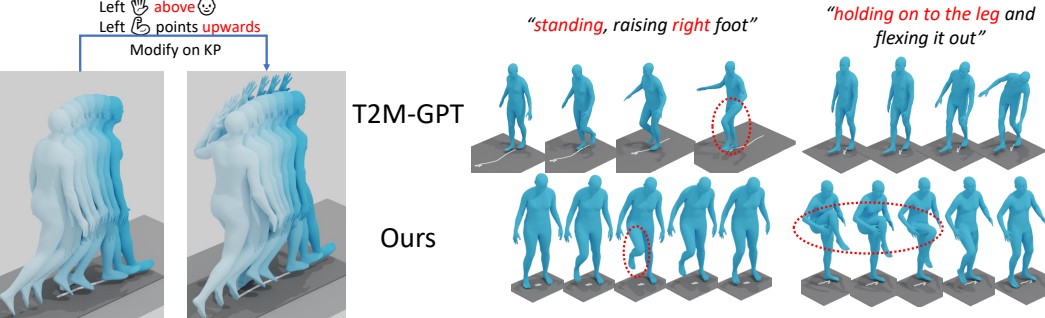

Figure 6: Our model supports fine-grained modification on motion via modification on KP.

Figure 7: Visualization of generated samples. Compared to T2M-GPT, our method provides a better response to prompts with explicit constraints on specific body parts.

## 6.4 ABLATION STUDIES

Ablation study results on KPG are shown in Tab. 5.

**KP mediation.** By using our joint space without KP mediation, we still present a competitive result, showing the efficacy of motion-KP joint space.

**Direct mapping.** By directly mapping with no KP involved, we present a similar performance compared to previous methods. It demonstrates the significance of KP in conveying action semantics.

**Different KP sets.** We examine the contribution of different KP sets: joint KP (PP), joint pair KP (PRPP, PDP), limb KP (LAP, LOP), and body KP (GVP). A leave-one-out style evaluation shows the elimination of joint KP and joint pair KP results in notable performance degradation, while the influence of the rest is relatively subtle.

## 7 DISCUSSION

Here, we discuss the limitations and prospects of KP and KP-based applications. **First**, KP could be extended beyond its current criteria of sign. These criteria guarantee objectivity but overlook important kinematic information like movement amplitude and speed. Also, due to the granularity of the adopted skeleton, fine-grained kinematic information on fingers is not well-preserved. The exploration of amplitude/speed/finger-based KP would be a promising goal to pursue. **Second**, KPB could be extended to datasets with other modalities, like 2D pose and egocentric action datasets. Though these modalities provide incomplete 3D information, we could extract KP that is credibly accessible across modalities. **Third**, with the convenient conversion from KP to text, auxiliary text descriptions could be automatically generated for motions via KP. **Fourth**, KPG could be extended by paraphrasing existing prompts and combining different Phrases.

## 8 CONCLUSION

In this paper, we proposed an intermediate representation to bridge human motion and action semantics as the Kinematic Phrase. By focusing on objective kinematic facts of human motion, KP achieved proper abstraction, interpretability, and generality. A motion understanding system based on KP was proposed and proven effective in motion interpolation, modification, and generation. Moreover, a novel motion generation benchmark Kinematic Prompt Generation is proposed. We believe that KP has great potential for advancing motion understanding.

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

APPENDIX

## A   OVERVIEW

This supplementary material presents more details and additional results not included in the main paper due to page limitation. The list of items included are:

- More details about *kinematic phrase* in Sec. B.
- More details about *kinematic phrase base* in Sec. C.
- Our method details in Sec. D.
- More details about *kinematic prompt generation* in Sec. E.
- Additional experimental details in Sec. F.
- Video demo (separate file: *125.mp4* and *kpg_samples.mp4*).

## B   KINEMATIC PHRASE DETAILS

This section lists the details of the six defined types of KP. During extraction, the indicator is set as zero if it is smaller than 1e-4.

### B.1   POSITION PHRASE

There are 36 phrases, corresponding to 36 interested $\langle joint, \ reference \ vector \rangle$ pairs like $\langle left \ hand, \ forward \ vector \rangle$. It is composed of 14 joints (except the pelvis and both eyes) and 3 reference vectors with originally 42 Phrases. Since there are joints that define the reference vectors (e.g. hips for the leftward vector). Specifically, shoulders and hips are excluded for the leftward vector, and hips are excluded for the upward vector, resulting in 42-6=36 PPs. The pairs are listed in the file `KP/pp.txt`.

### B.2   PAIRWISE RELATIVE POSITION PHRASE

There are 242 phrases corresponding to 242 interested $\langle joint, \ joint, reference \ vector \rangle$ triplets like $\langle left \ hand, \ right \ hand, forward \ vector \rangle$, listed in the file `KP/prpp.txt`. It is composed of 136 joint pairs and 3 reference vectors with originally 408 Phrases. 24 joint pairs that are linked by limbs are filtered out. 30 Eye-related pairs are filtered out except (the left eye, and right eye) due to the others are covered by head-related pairs. 4 triplets that the relationship barely changes along the reference vector (e.g., right knee, left hip, and leftward vector) are filtered out. These result in 408-24*3-30*3-4=242 PRPPs.

### B.3   PAIRWISE DISTANCE PHRASE

Joint pairs that are connected by human body topology are filtered out, like hand-elbow and shoulder-hip. There are 81 phrases corresponding to 81 interested $\langle joint, \ joint \rangle$ pairs like $\langle left \ hand, \ right \ hand \rangle$, listed in the file `KP/pdp.txt`. It is composed of 105 joint pairs (except both eyes). 24 pairs that are linked by the human body are filtered out, resulting in 81 PDPs.

### B.4   LIMB ANGLE PHRASE

There are 8 phrases corresponding to 8 interested limbs, listed in the file `KP/lap.txt`. The arms, legs, and their link with the upper body are included as 8 LAPs.

### B.5   LIMB ORIENTATION PHRASE

There are 33 phrases corresponding to 33 interested $\langle limb, \ reference \ vector \rangle$ pairs like $\langle left \ shank, \ right \ vector \rangle$, listed in the file `KP/lop.txt`. The 8 arm and leg limbs, head, collarbones, hips, torsos, and upper-body, paired with the 3 reference vectors, result in 45 LOPs in

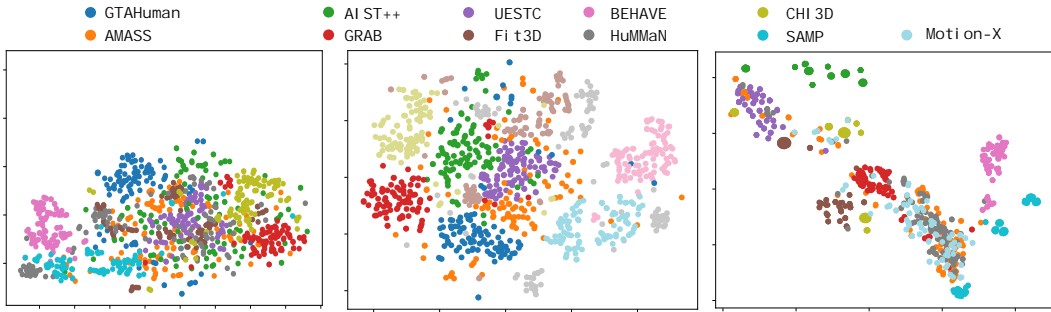

Figure 8: Motion, KP, and text distribution of Kinematic Phrase Base.

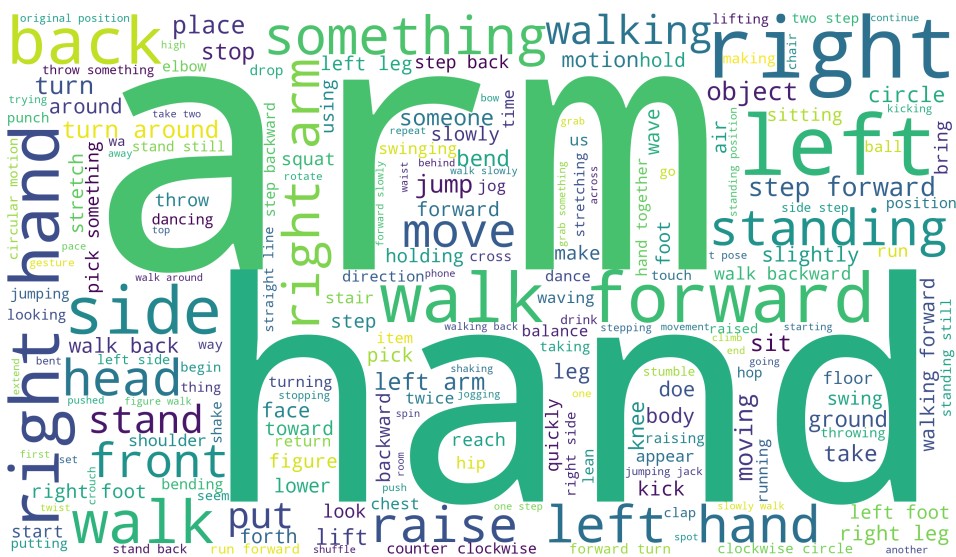

Figure 9: Word cloud visualization of the texts in Kinematic Phrase Base.

total. 12 pairs that barely change along the reference vector (e.g., left-right hip, leftward vector) are filtered out, resulting in 45-12=33 LOPs.

### B.6 GLOBAL VELOCITY PHRASE

There are 3 phrases corresponding to the velocity direction with respect to the three reference vectors.

## C KINEMATIC PHRASE BASE DETAILS

Over **140 K** motion sequences are collected to construct the Kinematic Phrase Base, including **9 M** frames (in 30 FPS) with **48 K** different sentences, covering a vocabulary size of **7,418**. Here, we illustrate the distribution of the collected database represented in motion, KP, and text in Fig. 8. Besides, a word cloud visualization of the texts in the database is illustrated in Fig. 9.

## D METHOD DETAILS

### D.1 LOSSES FOR JOINT SPACE LEARNING

Reconstruction loss $\mathcal{L}_{rec}$ compares the GT with the outputs of the VAEs. L1 losses are calculated for the motion representation $M, \hat{M}$, KP $C, \hat{C}$, the skeleton joints $J, \hat{J}$, the down-sampled mesh

vertices $V, \hat{V}$, and the joint accelerations $A, \hat{A}$.

$$\mathcal{L}_{rec} = \sum_{\cdot \in \{m,p,mp\}} ||M. - \hat{M}.||_1 + ||C. - \hat{C}.||_1 + ||J. - \hat{J}.||_1 + ||V. - \hat{V}.||_1 + ||A. - \hat{A}.||_1. \quad (4)$$

KL divergence loss $\mathcal{L}_{KL}$ encourages each distribution to be similar to a normal distribution $\pi = \mathcal{G}(0, I)$ by minimizing the Kullback-Leibler(KL) divergence between the normal distribution and the learned motion and KP distributions. The loss is calculated as

$$\mathcal{L}_{KL} = KL(\phi_m, \pi) + KL(\phi_p, \pi). \quad (5)$$

Distribution alignment loss $\mathcal{L}_{da}$ encourages the distributions of motion and KP to resemble each other by minimizing the KL divergence between them. The loss is calculated as

$$\mathcal{L}_{da} = KL(\phi_m, \phi_p) + KL(\phi_p, \phi_m). \quad (6)$$

Embedding alignment loss $\mathcal{L}_{emb}$ encourages the sampled latent vectors to be aligned by minimizing their L1 distance. The loss is calculated as

$$\mathcal{L}_{emb} = ||z_m - z_p||_1. \quad (7)$$

The overall loss is calculated as

$$\mathcal{L} = \lambda_1 \mathcal{L}_{rec} + \lambda_2 \mathcal{L}_{KL} + \lambda_3 \mathcal{L}_{da} + \lambda_4 \mathcal{L}_{emb}, \quad (8)$$

where $\{\lambda_i\}_{i=1}^4$ are weighting coefficients.

# E    KINEMATIC PROMPT GENERATION DETAILS

## E.1    PROMPTS

We provide the 840 text prompts converted from KP in the file `kpg.txt`.

## E.2    DETAILS OF ACCURACY COMPUTATION

Given the generated motion X with t frames and the target KP $c_i \in \{-1, 0, 1\}$, we first extract KP sequence corresponding to $c_i$ as $C_i \in \{-1, 0, 1\}^t$. Then, $c_i \in C_i$ is recognized when there is the sequence $c_i, c_i, c_i, c_i, c_i$ is a consecutive subsequence of $C_i$. For PRPP and LOP evaluation, the sequence $c_i^0, c_i^0, c_i^0, c_i^0, c_i^0, c_i^1, c_i^1, c_i^1, c_i^1, c_i^1$ is the expected subsequence.

# F    EXPERIMENT DETAILS

## F.1    IMPLEMENTATION DETAILS

The Motion VAE and KP VAE share the same structure: a 4-layer transformer encoder, a 4-layer transformer decoder, and a fully connected layer for final outputs. The denoiser adopted for text-to-motion is designed as a 4-layer transformer decoder. The latent size is set to 256. $\{\lambda_i\}_{i=1}^4$ are set as 1. The learning rate is decayed at 4,000 epochs for joint space training and at 2,000 epochs for text-to-motion latent diffusion model training.

## F.2    MOTION GENERATION SETTINGS

For HumanML3D (Guo et al., 2022a), motion sequences are generated for 10 seconds given a text prompt. For KPG, the models are required to generate 120 frames given a text prompt.

R-Precision is calculated in a similar way to Guo et al. (2022a). For each generated motion, its text description is mixed with 31 randomly selected mismatched descriptions from the test set. The cosine distances between the motion feature and text features are computed. The average accuracy at the top-1 place is reported.

FID is adopted to measure the divergence between the GT motion distribution and the generated motion distribution in the latent space.

| Methods | Top-1 Acc.%↑ | Top-5 Acc.%↑ |
|---|---|---|
| MotionCLIP Tevet et al. (2022a)* | 40.90 | 57.71 |
| 2s-AGCN Shi et al. (2019); Punnakkal et al. (2021)* | 41.14 | 73.18 |
| Ours* | 45.73↑$_{4.6(11.2\%)}$ | 76.92↑$_{3.7(5.1\%)}$ |
| Ours | **47.36**↑$_{6.2(15.1\%)}$ | **78.44**↑$_{5.3(7.2\%)}$ |

Table 6: Motion classification results on BABEL-60. * indicates the model is trained with the BABEL train set only.

| Method | MDM (Tevet et al., 2022b) | MLD (Chen et al., 2023) | T2M-GPT (Zhang et al., 2023a) | Ours |
|---|---|---|---|---|
| #params | 23M | 42.7M | 228M | 45.1M |

Table 7: Model Size Comparison.

Diversity measures the variance of the generated motion sequences. It is calculated as the average latent distance between two randomly sampled generated motion sets. The set size is set as 300 in this paper.

Multimodality measures the variance of the generated motion sequences within each text prompt. For each description, two subsets of motion sequences with the same size are generated, and then the Multimodality is calculated as the average distance between the two sets of motions in the latent space. The size of each subset is set as 10 in this paper.

### F.3 KP for Action Recognition

We follow the BABEL 60-classes benchmark Punnakkal et al. (2021) and report the Top-1 accuracy and Top-5 accuracy on the BABEL validation set. A simple MLP is adopted to compute the prediction scores from the encoded KP embeddings. As shown in Tab. 6, we achieve superior performance on BABEL-60 compared to previous SOTAs on Top-1 accuracy, proving the efficacy of our KP-mediated motion classification.

### F.4 Model Size comparison.

We compare the number of parameters in our model and previous SOTAs in Tab. 7. As shown, with a model size comparable to MLD (Chen et al., 2023) and significantly lower than T2M-GPT (Zhang et al., 2023a), we achieve competitive performance on conventional benchmarks and even better performance with the newly proposed KPG.

### F.5 User Study Details

#### F.5.1 User Study Design

As stated in the main text, we adopt a direct Q&A-style user study instead of a popular preference test or ratings. Here we clarify the reason for this design choice. First, this design is more suitable in evaluating **semantic consistency**, which we identify as categorical instead of continuous at the sample level. That is, it is hard to tell whether a motion is more `raising left-hand up` than another. Instead, there is only whether a motion is `raising left-hand up` or not. Therefore, we chose to present a direct question on semantic consistency. Second, this design explicitly decouples the evaluation of text-to-motion into semantic consistency and naturalness, corresponding to R-Precision and FID. When rating motions or choosing between two motions, it is hard to guarantee the users make choices according to the expected standard. Therefore, we explicitly ask decoupled binary questions for decomposition. Third, it helps reduce annotation costs. For preference testing, the complexity is $O(N^2)$, while with our user-study protocol, the complexity is only $O(N)$. In consideration of our primary focus on semantic consistency, we adopt this protocol. We also admit this protocol is sub-optimal in naturalness evaluation, which is a continuous factor. We present the results on naturalness as a reference in the following sections.

| FID = 0.544 R-P@1 = 0.266 | | Semantic consistency | | | Sum |
|---|---|---|---|---|---|
| | | Yes | Partially | No | |
| Naturalness | Yes | 0.40 | 0.18 | 0.10 | 0.68 |
| | No | 0.03 | 0.11 | 0.18 | 0.32 |
| Sum | | 0.43 | 0.29 | 0.28 | 1 |

(a) MDM (Tevet et al., 2022b).

| FID = 0.212 R-P@1 = 0.473 | | Semantic consistency | | | Sum |
|---|---|---|---|---|---|
| | | Yes | Partially | No | |
| Naturalness | Yes | 0.34 | 0.13 | 0.04 | 0.51 |
| | No | 0.10 | 0.14 | 0.25 | 0.49 |
| Sum | | 0.44 | 0.27 | 0.29 | 1 |

(b) MLD (Chen et al., 2023).

| FID = 0.141 R-P@1 = 0.292 | | Semantic consistency | | | Sum |
|---|---|---|---|---|---|
| | | Yes | Partially | No | |
| Naturalness | Yes | 0.50 | 0.16 | 0.05 | 0.71 |
| | No | 0.06 | 0.08 | 0.15 | 0.29 |
| Sum | | 0.56 | 0.24 | 0.20 | 1 |

(c) T2M-GPT (Zhang et al., 2023a).

| FID = 0.631 R-P@1 = 0.274 | | Semantic consistency | | | Sum |
|---|---|---|---|---|---|
| | | Yes | Partially | No | |
| Naturalness | Yes | 0.52 | 0.21 | 0.02 | **0.75** |
| | No | 0.05 | 0.06 | 0.14 | 0.25 |
| Sum | | **0.57** | **0.27** | 0.16 | 1 |

(d) Ours.

Table 8: Detailed user study results on HumanML3D.

| Accuracy = 50% | | Semantic consistency | | | Sum |
|---|---|---|---|---|---|
| | | Yes | Partially | No | |
| Naturalness | Yes | 0.29 | 0.09 | 0.53 | 0.91 |
| | No | 0.04 | 0.01 | 0.04 | 0.09 |
| Sum | | 0.33 | **0.10** | 0.57 | 1 |

(a) T2M-GPT Zhang et al. (2023a).

| Accuracy = **54%** | | Semantic consistency | | | Sum |
|---|---|---|---|---|---|
| | | Yes | Partially | No | |
| Naturalness | Yes | 0.33 | 0.07 | 0.51 | **0.92** |
| | No | 0.04 | 0.01 | 0.04 | 0.08 |
| Sum | | **0.37** | 0.08 | 0.55 | 1 |

(b) Ours.

Table 9: Detailed user study results on KPG.

### F.5.2 USER STUDY ON CONVENTIONAL TEXT-TO-MOTION

Detailed user study results on HumanML3D are demonstrated in Tab. 8. As shown, both FID and R-P@1 are not totally consistent with the user reviews, indicating these black-box-based metrics might be sub-optimal for motion generation evaluation. Meanwhile, the four evaluated methods present a similar positive correlation between semantic consistency and naturalness. Moreover, it shows that generating natural motions is a little harder than generating partially semantic-consistent motions, which might be a potential direction to advance motion generation.

### F.5.3 USER STUDY ON KPG

Detailed user study results on KPG are demonstrated in Tab. 9. Our proposed Accuracy shares a similar trend with user-reviewed semantic consistency between the two methods. Both methods receive good naturalness reviews, which could result from the simple prompt structure of KPG.

Furthermore, we provide detailed consistency statistics between KP-inferred Accuracy and user-reviewed semantic consistency in Tab. 10. Samples generated from T2M-GPT and our method are included. KP and users provide similar reviews for over 80% of the samples, showing good consistency. With respect to user reviews, KP-inferred Accuracy has a higher false positive rate (0.12 / 0.52 = 0.2308) than a false negative rate (0.04 / 0.48 = 0.0833). We find there are two typical false positive scenarios. First, the generated motion results in rather small indicators, close to the 1e-4 threshold. KP captures this, however, it is hard for humans to notice such subtle movements. Second, as shown in Fig. 10, the generated motions sometimes tend to be redundant compared to the given prompts. Users might be distracted, overlooking the targeted semantics. We find this happens more for T2M-GPT generated samples (in Fig. 7, extra walking motion; in Fig. 10, extra right-hand waving motion), while our method manages to provide more concise responses. We think this could partially explain the higher Diversity of T2M-GPT in Tab. 4. For the first scenario, we think an adaptive threshold w.r.t. the overall motion intensity would be helpful, since to human perception, the relative amplitude is usually more important than the absolute amplitude. Also, as stated in Sec. 7, extending KP to amplitude might also help. The second scenario urges us to rethink the current text-to-motion task setting. For a "matched" motion-text pair, should the text semantics be a subset of motion semantics, or strictly match? Also, is it expected to increase diversity by introducing redundant motions? We identify these questions as interesting points of attack and leave them for future exploration.

|  |  | User Reviewed | | | Sum |
| --- | --- | --- | --- | --- | --- |
|  |  | Yes | Partially | No |  |
| KP-Inferred | Yes | 0.32 | 0.08 | 0.12 | 0.52 |
|  | No | 0.03 | 0.01 | 0.44 | 0.48 |
| Sum |  | 0.35 | 0.09 | 0.56 | 1 |

Table 10: Detailed consistency statistics between KP-inferred Accuracy and user-reviewed semantic consistency.

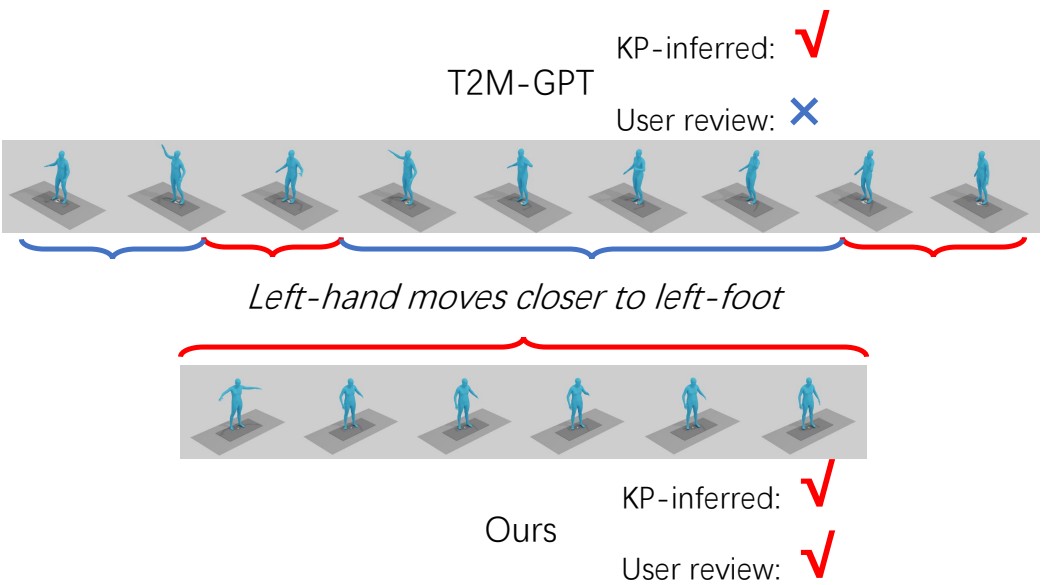

Figure 10: For KPG, we generate more concise motion than T2M-GPT (Zhang et al., 2023a).

| Method | PP % | PRPP % | PDP % | LAP % | LOP % | GVP % | Overall |
| --- | --- | --- | --- | --- | --- | --- | --- |
| MDM Tevet et al. (2022b) | **100.00** | 21.24 | **100.00** | **100.00** | 12.12 | **100.00** | 44.40 |
| MLD Chen et al. (2023) | **100.00** | 21.43 | **100.00** | **100.00** | 15.15 | **100.00** | 44.76 |
| T2M-GPT Zhang et al. (2023a) | **100.00** | 26.45 | 98.77 | 87.50 | 19.70 | 83.33 | 47.86 |
| Ours | **100.00** | **32.05** | **100.00** | **100.00** | **24.24** | **100.00** | **52.14** |

Table 11: Detailed results of KPG with respect to different KPs.

## F.6 FAILURE ANALYSIS ON KPG

Detailed results of KPG with respect to different types of KPs are demonstrated in Tab. 11. As shown, most methods provide accurate motions for the relatively simpler PP, PDP, LAP, and GVP. However, T2M-GPT Zhang et al. (2023a) occasionally failed on some rather simple cases. In contrast, the major challenge is the PRPP and LOP-related prompts. We identify two reasons. First, the temporal composition of KPs increases the difficulty. Second, some prompts are easy for humans but hard for current models, as would be shown in the following.

With the attached video `rebuttal.mp4`, we further demonstrate more failure cases on KPG. For accuracy, a major failure mode is due to the cases like "right shoulder is in front of left foot, then behind left foot". These are simple body joint relations like exercising instructions, however, not usually explicitly described in general datasets. This reveals that current models tend to be suboptimal in real understanding of the human body structure. Also, previous efforts might even corrupt these. For user study, there are two major failure modes. First, our model could sometimes generate a motion with limited amplitude for target KPs, which could be captured by KP-based accuracy but humans could lose track of them. Second, previous models like T2M-GPT generate redundant motions for simple prompts, which could confuse users.

### F.6.1 MORE VISUALIZATIONS

Some qualitative results of KPG are included in the video `kpg_samples.mp4`. More visualizations are included in the video `125.mp4`.

