# OpenReview forum: "BRIDGING THE GAP BETWEEN HUMAN MOTION AND ACTION SEMANTICS VIA KINEMATIC PHRASES"
_ICLR.cc/2024/Conference — Submitted to ICLR 2024_

### Official Review · Reviewer_DPfm · 2023-10-30

**Soundness:** 3 good
**Presentation:** 3 good
**Contribution:** 3 good
**Rating:** 6
**Confidence:** 4

**Summary:**

This paper proposes to use kinematic phrases as a quantifiable way of judging human motion. Kinematic phrases are defined as objective facts based on the state and relationship between human body parts. A VAE-based model for KP is learned to align motion latent space with KP latent space, which is then used for tasks such as motion interpolation, modification, and generation. Experiments show that using KP as a representation aids in objective motion understanding while using KP as a criterion serves as a better way to judge motion generation quality.

**Strengths:**

- I find the formulation of KP as a motion representation refreshing and intuitive. It provides a quantifiable way to judge motion and serves as a much-needed metric for human motion generation. While the community has made significant progress, it appears that the current metrics are not quite indicative of the models' performances.
    - The design of KP seems well-principled and well-thought-out. Using indicator signs to show the rough trend of the motion and compressing a continuous space of kinematic motion to a 403-bit value could be a powerful way to compress the "essence" of motion.
    - KP provides a very quantifiable way to connect high-level semantics with low-level motion facts. It can serve as the bridge between kinematic motion sequences and high-level human descriptions and commands.
    - The result on KPG (Table 2) shows that the current human motion generation method falls far short in generating accurate semantically correct human motion. A ~50% success rate shows that current popular methods, while they can generate high-quality motion, lack a deeper understanding of phrases and motion.
- The proposed KP-guided motion generation method achieves state-of-the-art results in motion interpolation and generation.

**Weaknesses:**

- I find the lack of more analysis on current models and their failure modes on the KPG benchmark a missed opportunity. What are the main failure modes these methods are failing on and which part is the proposed method winning at? The analysis could provide far more valuable insight into the current human motion generation community, while the FID and diversity tell very little.
- More qualitative result on using the proposed KP and motion VAE method is needed for better assessment of the proposed method. The provided results are very short and not super informative. No qualitative result is provided for the motion interpolation and modification tasks.

**Questions:**

- Are all the models (MDM/MotionGPT/etc.) trained with the same data? Is the KP-guided joint latent space trained on all the data in Table 1 or similar to prior methods?
- I do find the proposed method the weaker link in the submission. Not enough information and intuition are provided for the proposed method and no supplemental materials are provided. How is the alignment done? How does $D_m$ and $D_p$ take an arbitrary combination of $z_m$ and $z_p$ as input? How is the interpolation done based on the output latent codes?
- I am willing to raise my score if more analysis on thee current models and their failure modes on the KPG benchmark could be provided, as well as some more insight into the qualitative performance of the proposed method.

---

> ### Author Response · Authors · 2023-11-20
>
> Dear Reviewer DPfm:
>
> we extend our deepest gratitude for your time, efforts, thorough review, and constructive advice in improving the paper. We sincerely appreciate for recognizing KP designation as well-principled and well-thought-out. We sincerely for recognizing our work as well-presented. Herein, we respond to each of your concerns.
>
> 1. **Analysis of failure modes.** In Appendix F.3.3 and Figure 10, we include a typical user-studied KPG failure case of current models. where redundant motion is generated. To demonstrate more, please refer to the general response part 2.
>
> 2. **More qualitative results.** Please refer to the general responses.
>
> 3. **Training data.** In Tab. 3, all methods with * are trained on HumanML3D only, including the KP-guided latent space. The rest is trained with KPB.
>
> 4. **Details of the adopted method.** Our model is mostly designed for a fast and lightweight effectiveness evaluation of KP.
> The results show that even by simply incorporating KP with a simple VAE structure, considerable improvements are achieved.
>
> 5. **Alignment.** For alignment, the losses in Eq. 4 and Eq. 6-7 are designed to align the motion latent space with the KP latent space.
>
> 6. **Decoder inputs.** During training, as the reconstruction loss in Eq. 4, an arbitrary combination of $z_m$ and $z_p$ are sent into the decoders, resulting in the flexible input of the decoders.
>
> 7. **Interpolation details.** Sorry for the potential ambiguity. The interpolation is not conducted directly via manipulation of the latent codes. Instead, we extract latent code from the corrupted motion and KP sequences. With self-supervised learning, our model is capable of reconstructing the complete KP sequence from the latent code extracted from the corrupted KP sequence. With the reconstructed KP sequence, we further feed it into the encoder for a latent with complete information, then reconstruct the whole motion with the motion decoder.
>
> Thank you again for your constructive feedback. We will incorporate the corresponding modifications and expansions in the revision. In addition, the corresponding code and model weights will be open-source to ensure replication. If there are any further questions, please inform us. We would be very happy to do anything we can that would be helpful in the time remaining. Thanks!

---

> > ### Comment · Reviewer_DPfm · 2023-11-22
> > **Reviewer Response to Author Response**
> >
> > Thanks a ton for the detailed response!
> >
> > 1. Analysis of failure modes: I am referring more to how the current mode fails on the KPG task in general rather than the discrepancy between KPG and use reviewers. Which KPG's criterion does the current method often fail in? Is it the relative positions? Angles? Velocities?
> > 2. If "fast and lightweight "effectiveness evaluation" is the intended use case, then a similar bench mark method should be used together for a fair comparison. As other reviewers have mentioned, one of the biggest issues is the proposed method is not achieving SOTA. Also, consider including concrete model sizes in the table for a better picture.
> > 3. I find the provided interpolation video quite confusing. Is the red target given as the last frame target? Is this essentially motion inpainting/infilling?
> >
> > Overall, I find the proposed KP and KPG a positive contribution, but some issues remain with the proposed method and evaluation process. I feel like maybe spending more time on the proposed metric and the problems it reveals on the current method could be more important than proposing a new method.

---

> > > ### Author Response · Authors · 2023-11-22
> > >
> > > Dear Reviewer DPfm:
> > >
> > > Thanks very much for your timely response! We will address each of the concerns in the following.
> > >
> > > 1. **Analysis of failure modes**. Thanks a lot for your advice. We further provide detailed KPG results as shown in the following table. Most methods provide accurate motions for the relatively simpler PP, PDP, LAP, and GVP. However, T2M-GPT occasionally failed in some rather simple cases. The major challenge is the PRPP and LOP-related prompts. We identify two reasons. First, the temporal composition of KPs increases the difficulty. Second, some prompts are easy for humans but hard for current models, like the visualized "right shoulder is in front of left foot, then behind left foot". The corresponding table and descriptions are updated in Appendix F.6.
> > > | Method | PP         | PRPP      | PDP        | LAP        | LOP       | GVP        | Overall   |
> > > |---------|------------|-----------|------------|------------|-----------|------------|-----------|
> > > | MDM     | **100.00** | 21.24     | **100.00** | **100.00** | 12.12     | **100.00** | 44.40     |
> > > | MLD     | **100.00** | 21.43     | **100.00** | **100.00** | 15.15     | **100.00** | 44.76     |
> > > | T2M-GPT | **100.00** | 26.45     | 98.77      | 87.50      | 19.70     | 83.33      | 47.86     |
> > > | Ours    | **100.00** | **32.05** | **100.00** | **100.00** | **24.24** | **100.00** | **52.14** |
> > >
> > > 2. **Fair comparison**. Thanks for your valuable advice! We compare the number of parameters in the following table. With a model size comparable to MLD and significantly lower than T2M-GPT, we achieve competitive performance on conventional benchmarks and even better performance with the newly proposed KPG. Despite the proposed method not thoroughly achieving SOTA on conventional metrics, we achieve SOTA in the scope of KPG, which is more reliable in reasonably analyzing specific motion generation ability as shown in the General Response part 1. The corresponding table and descriptions are updated in Appendix F.4.
> > > | Method | MDM | MLD   | T2M-GPT | Ours  |
> > > |--------|-----|-------|---------|-------|
> > > | #param | 23M | 42.7M | 228M    | 45.1M |
> > >
> > > 3. **Motion Interpolation Sample**. Yes, the red target is the last frame target. The given sample is more difficult than reported for quantitative evaluation (randomly dropping 50% frames), showing our model could handle both short-term interpolation (Tab.3) and longer-term interpolation/inpainting/infilling.
> > >
> > > Thank you again for your constructive feedback. If there are any further questions, please inform us. We would be very happy to do anything we can that would be helpful in the time remaining. Thanks!

---

> > > > ### Comment · Reviewer_DPfm · 2023-11-23
> > > > **Response to Followup Questions**
> > > >
> > > > Thanks a ton for making analysis in such a short time.
> > > >
> > > > I am surprised that the 100% success rate over some of the metrics like "position phrase", "Pairwise Distance Phrase", and "Limb  Angle  Phrase", meaning the SOTA methods can already generate motion that match these all the time?

---

> ### Author Response · Authors · 2023-11-23
>
> Thanks for your feedback!
>
> With the current setting of KPG, most methods barely fail for these phrases, given the simplicity of these Phrases (e.g., "left-hand moves forwards" for position phrases, "left-hand moves closer to the right hand" for position distance phrases, "left arm bends" for limb angle phrases). The descriptions corresponding to these phrases are also covered in datasets like HumanML3D, facilitating the success of these.
>
> However, the discrepancy between the successful KPs and the others is noticeable. For example, Position Phrases and Pairwise Relative Position Phrases could be executed with similar motions, like "left-hand moving forwards" and "left-hand is first behind the pelvis, then in front of the pelvis". However, the discrepancy shows the current models are unable to learn such reasoning ability.
>
> Also, please notice the setting in Appendix F.2. For each prompt in KPG, the models would generate 120 frames of motion, while the target  KP is only required to exist for 5 consecutive frames. This temporal constraint is rather loose considering that the phrases do not need such a long time to execute for a human. Initially, we tried a more strict setting, where only 10-frame motions are generated. All methods fail to provide feasible results. Considering our current focus on basic semantic consistency, we switch to the loose setting and leave the exploration of temporal constraint for future work.
>
> Please let us know if there is anything helpful we can do in the remaining time.

---

### Official Review · Reviewer_HZyv · 2023-10-31

**Soundness:** 2 fair
**Presentation:** 3 good
**Contribution:** 2 fair
**Rating:** 6
**Confidence:** 3

**Summary:**

This work proposes an intermediate human motion representation based on hand-crafted features with the goal to bridge the gap between human motion and action descriptions. This representation can be used for various tasks such as motion interpolation and motion generation.

**Strengths:**

The authors propose a deterministic and easy to replicate approach to extract low-level features from human motion. The features are generally well-described in Section 3.1 and Figure 2 aids in understanding.

**Weaknesses:**

The claim that “Kinematic Phrases” bridges the gap between human motion and action is over-selling the paper: one could imagine a person either (1) slightly hammering (slight up-down motion of wrist) or (2) swiping through a book (slight left-right motion of wrist) while standing perfectly still and the proposed KP would not pick up the difference due to the small motion - this has also been acknowledged by the authors in the Discussion. The advantages of textual descriptions of motion or of action labels is that they are easy for humans to understand: while KPs are easier to understand that directly looking at joint angles they are much more difficult to interpret than language and generating novel and sensible KPs from scratch sounds difficult for a human to do without tools.


Some information is missing, for example: In 3.1. the authors define various sets of Kinematic Phrases (36 PPs, 242 PRPPs, 81 PDPs, etc) - where do those numbers come from?

KP as evaluation alternative to common methods such as FID is also questionable: Even broken and unrealistic human motion would produce “valid” KP that would be difficult to differentiate from valid sequences.

I strongly suggest the Authors to use Equations on separate lines when describing the models in 4.2.

Minor issues:
* On Page 3 $r^f = r^r \times r^r$ should be $r^f = r^r \times r^u$
* It is a bit unclear in 3.1 what the “Phrase” is exactly: is it the sign or is it the signal

**Questions:**

* What is the “human cognitive view” on Page 3?
* It seems that the authors approach could be useful for approaches such as nearest neighbor: did the authors experiment with those approaches i.e. for action recognition or motion to text?

**REBUTTAL END**

While the rebuttal answers most of my questions I am still concerned about the “bridging the gap” part that I feel is not adequately answered:

1:  The claim of "bridging the gap”: On the one hand the authors say that KPs are automatically extracted and that the “major burden of understanding KP is left to the machines”, on the other hand they maintain that KPs are “easily” interpretable - which I still disagree with: For example, in Figure 3 I would not be able to “easily” understand only from the KP what the poses would be. To me interpretability is made difficult due to two reasons: (1) the relatively large number of phrases and hierarchies (2) the aggressive discretization

However, I also believe that a deterministic and easy to replicate approach to extract low-level features from human motion is a valuable contribution for other downstream tasks (i.e. quality measures) which is still an open problem for human motion.

If the authors tone down their interpretability claims a bit and more importantly present the same level of detail in Chapter 4.2 as they did in the previous chapters, i.e. describing the method in Equations and not just in text I am willing to raise my rating.

---

> ### Author Response · Authors · 2023-11-20
>
> Dear Reviewer HZyv:
>
> First and foremost, we extend our gratitude for your time and efforts on the review, as well as the helpful advice in improving the paper. We sincerely for recognizing our work as well-presented. Herein, we respond to each of your concerns.
>
> 1. **The claim of "bridging the gap"**. First, we believe that we should not let the perfect be the enemy of the good. Despite the limited granularity in the current version, it could be easily extended with the proposed framework. Second, we argue that the target of KP is not fully replacing language descriptions for action semantics representation. In essence, KP is designed as an **automatic symbolization system** for motion. We expect KP to become an interface for motion-semantics mapping, where the major burden of understanding KP is left to the machines, while humans could interfere with the process as assistance, given the easily interpretable atomic Phrases. Thus, humans are not required to be able to generate KP sequences from scratch.
>
> 2. **The number of Phrases**. Here we list the details for KP filtering. These will be updated in the appendix.
>     - Position Phrase (PP). It is composed of 14 joints (except the pelvis and both eyes) and 3 reference vectors with originally 42 Phrases. Since there are joints that define the reference vectors (e.g. hips for the leftward vector). Specifically, shoulders and hips are excluded for the leftward vector, and hips are excluded for the upward vector, resulting in 42-6=36 PPs.
>     - Pairwise Relative Position Phrase (PRPP). It is composed of 136 joint pairs and 3 reference vectors with originally 408 Phrases. 24 joint pairs that are linked by limbs are filtered out. 30 Eye-related pairs are filtered out except (for the left eye, and right eye) due to the others are covered by head-related pairs. 4 triplets that the relationship barely changes along the reference vector (e.g., right knee, left hip, and leftward vector) are filtered out. These result in 408-24*3-30*3-4=242 PRPPs.
>     - Pairwise Distance Phrase (PDP). It is composed of 105 joint pairs (except for both eyes). 24 pairs that are linked by the human body are filtered out, resulting in 81 PDPs.
>     - Limb Angle Phrase (LAP). The arms, legs, and their link with the upper body are included as 8 LAPs.
>     - Limb Orientation Phrase (LOP). The 8 arm and leg limbs, head, collarbones, hips, torsos, and upper body, paired with the 3 reference vectors, result in 45 LOPs in total. 12 pairs that barely change along the reference vector (e.g., left-right hip, leftward vector) are filtered out, resulting in 45-12=33 LOPs.
>
> 3. **KPG evaluation of naturalness**.  We agree that naturalness is necessary, and also provide user-study results on naturalness in the supplementary materials. However, KP-based accuracy is not an all-in-one metric. With its main target as semantic consistency evaluation, it is more an alternative to R-Precision instead of FID. We believe pursuing an interpretable naturalness metric like KP-based accuracy for semantic consistency is worth exploring.
>
> 4. **Equations**. Thanks for your advice. We will incorporate these in the revision.
>
> 5. **Typos.** Thanks for pointing out. We will fix it.
>
> 6. **Phrase.** Sorry for the ambiguity. By Phrase, we mean the signs as $[-1, 0, 1]$. We will clarify this in the revised version.
>
> 7. **Human cognitive view.** It means that the reference vectors are derived from human cognitive references for egocentric descriptions of actions. For example, we are more interested in which direction vectors humans adopt when saying "My hand is moving leftwards", and find the vector from right hip to left hip.
>
> 8. **KP for action recognition.** Thanks for your constructive feedback. We conduct a simple experiment for action recognition on the BABEL validation set, where promising results are shown with Top-1 accuracy improving from 41.14 to over 45. The details are updated in the supplementary materials.
>
> Thank you again for your helpful feedback. We will incorporate the corresponding modifications and expansions in the revision. In addition, the corresponding code and model weights will be open-source to ensure replication. If there are any further questions, please inform us. We would be very happy to do anything we can that would be helpful in the time remaining. Thanks!

---

### Official Review · Reviewer_537a · 2023-11-01

**Soundness:** 3 good
**Presentation:** 3 good
**Contribution:** 3 good
**Rating:** 6
**Confidence:** 4

**Summary:**

This paper presents Kinematic Phrases, a set of motion features extracted based on manually defined rules, which is claimed to enhance motion interpretability to bridge the gap and tackle the many-to-many problem between human motion and action semantics. Based on KP, the paper introduces KPG as a benchmark to evaluate whether a generated motion is consistent with particular action semantics.

**Strengths:**

The introduction to this paper is well thought out and motivated. Modeling many-to-many mappings between motion and semantics is challenging and interpretability has not been well addressed in this area.

The evaluation results show the merits of the proposed approach, but weakness is also revealed (see below).

**Weaknesses:**

Considering that the proposed method does not perform as well as the diffusion-based baseline method on most of the standard metrics, the method also only slightly outperforms the baseline method in the user study by more than between 0.01 and 0.04, which means that on average maybe just 5 out of 200 sequences show better quality in terms of motion semantics. This result may not be strong enough to support the paper.

I would also suggest that the paper provide some qualitative results with their corresponding standard metrics values to show that the paper's methodology generates movements that are indeed more semantically correct, whereas the metrics show the opposite. The KPG could be used and the accuracy compared to the above results to provide further evidence.

**Questions:**

I wonder how to make a motion modification exactly, given another description (as shown in Figure 6), how to modify that phrase to another phrase corresponding to the description. In Figure 3,  seems some frames of phrase C are just masked out.

I think the description of KPG evaluation in section 5 is a bit unclear. what exactly is c_i and how is c_i \in C_i calculated. For example, it would be good to list their shapes

How KP handles ambiguity in the time dimension. While the discussion mentions amplitude and velocity constraints. For example, "the left eye is in front of the right eye and then behind the right eye", this would actually lead to a completely different phrase

Overall, the author's response to the concerns in the Weaknesses and questions is needed to make the final decision. I am happy to increase the rating if my concerns are addressed.

post-rebuttal:

Thank you for your comprehensive response, which largely addresses my concerns.

Regarding the temporal ambiguity: how does KPG accurately assess the correctness of a motion sequence when e.g., the second atomic action occurs significantly later, unlike GT? As I understand it, KPG would assign a low score in this scenario, but this may still be correct if the description doesn't specify timing.

I believe the paper has merit, particularly in its contributions to interpretability and fine-grained semantics evaluation within the field. Therefore, I plan to increase my rating.

---

> ### Author Response · Authors · 2023-11-20
>
> Dear Reviewer 537a:
>
> First and foremost, we extend our deepest gratitude for your thorough review and helpful advice in improving the paper. We sincerely appreciate for recognizing our work as well thought out and motivated. Herein, we respond to each of your concerns.
>
> 1. **The marginal advantage over SOTA.**
> Notably, our text-to-motion model could be identified as improving the latent-based diffusion proposed in MLD with the newly proposed KP.
> This choice is made to conduct a fast and light-weight effectiveness evaluation of KP.
> Therefore, a proper comparison target should be MLD, upon which considerable advantages are shown in the user-study results and KPG.
> For T2M-GPT and MotionGPT, which are the current SOTAs, we would like to emphasize the different paradigms they adopted, as well as the model scale.
> Both of them tokenized motions into a codebook, coupled the codebook with text tokens, and then adopted a different pipeline as an auto-regressive decoder-only generative model, which resembles GPT.
> Also, they possess a considerably bigger model scale. For example, the 18-layer transformer is adopted by T2M-GPT for the text-to-motion task, however, in comparison, only a 4-layer denoiser and a 4-layer with a frozen CLIP text encoder are adopted in our method.
> Therefore, we would claim that despite the diverse paradigm and model scale, by incorporating KP into an MLD-like pipeline, we manage to achieve comparable performance with SOTAs, proving the efficacy of KP.
> Furthermore, given the motion quantization ability of KP as mentioned by Reviewer DPfm, also its effortless conversion to natural language, we are optimistic about incorporating it with T2M-GPT for future work.
>
> 2. **Qualitative results with the standard metric values.** Please refer to the general responses, part 1.
>
> 3. **Details of motion modification.** We are sorry for the oversimplified illustration in Fig. 3. Here we provide a more detailed figure at [**this link**](https://ibb.co/3YV1zct) to illustrate the motion modification process. The image is also updated in the supplementary materials as modify.png, which will also be incorporated in the PDF.
>
>
> 4. **Details of KPG evaluation.** Given the generated motion X with t frames and the target KP $c_i \in [-1, 0, 1]$, we first extract the KP sequence corresponding to $c_i$ as $C_i \in [-1,0,1]^t$. Then,  $c_i \in C_i$ is recognized when there is the sequence $c_i, c_i, c_i, c_i, c_i$ is a consecutive subsequence of $C_i$. For PRPP and LOP evaluation, the sequence $c_i^0, c_i^0, c_i^0, c_i^0, c_i^0, c_i^1, c_i^1, c_i^1, c_i^1, c_i^1$ is the expected subsequence. We will update the details in the PDF.
>
> 5. **Temporal ambiguity of KP.** Notably, the conversion from motion to KP could involve no temporally down-sampling. In this sense, KP would not introduce new temporal ambiguity upon the captured motion. Given the example "the left eye is in front of the right eye and then behind the right eye", if it is included in the raw motion, KP would faithfully reflect the corresponding information.
>
> Thank you again for your constructive feedback. We will incorporate the corresponding modifications and expansions in the revision. In addition, the corresponding code and model weights will be open-source to ensure replication. If there are any further questions, please inform us. We would be very happy to do anything we can that would be helpful in the time remaining. Thanks!

---

### Official Review · Reviewer_87dC · 2023-11-03

**Soundness:** 3 good
**Presentation:** 3 good
**Contribution:** 3 good
**Rating:** 6
**Confidence:** 2

**Summary:**

The objective of the work is to fill the gap between motion and action semantics by proposing Kinematic Phrases, and intermediate, interpretable representation that focuses on kinematic facts. The authors state they construct a unified large-scale motion knowledge base, and then learn with self-supervision a motion-KP joint latent space that is later used for different target tasks: (i) motion interpolation, (ii) motion modification, and (iii) generation. Furthermore, they propose a benchmark called Kinematic Prompts Generation.
The experimental evaluation is performed on a public dataset and comparison with alternative approaches are reported.

**Strengths:**

The paper addresses an important point: the gap between raw motion data and action semantics, which negatively affects the ability of automatic methods to evaluate the quality of their results. The approach is fairly motivated and a discussion on existing approaches that should help better place the proposed method is reported.
The experiments seem extensive and cover the different tasks the authors propose. An ablation study is also reported. Details on the training procedure are provided, to favour the reproducibility of the main results.

**Weaknesses:**

I do not find the reading particularly easy and clear. There are some parts that are very dense in technical details and/or that make reference to previous approaches. Indeed, the paper strongly relies on previous works and is not fully self-contained. While of course, this is understandable, I would suggest the authors be sure that at least the minimal information to understand and appreciate the work is included (just as an example: the meaning of the metrics in Tab. 3).

I am not sure I fully understood the placement with respect to the literature: in what sense does the proposed method advance the state-of-art?

I find that one of the main target tasks for this work, i.e. motion generation, should be better clarified from the very beginning, in the introduction (the impression that I have is that it becomes clear just after a while, by reading the next sections).

The authors state their task is motion understanding, but in my opinion, they should be more specific: all the examples refer to walking sequences as if there was a particular interest in them. Nevertheless, the datasets employed in the experimental evaluation seem more rich. Some examples of different motions might help better identify the setting and the problem of interest.

The figures are not always enough explanatory. For instance, while Figure 2 is very clear, from Fig. 1 I would expect to have a better focus on what the authors mean when they say that there is a significant gap between motion and action semantics, but it is not clear to me

**Questions:**

In addition to my comments above, I report here some more questions I have, hoping this may help to improve the readability and understanding of the method:

- "For objectivity and actuality, KP captures sign changes with minimal pre-defined standards" I am not sure I understand this statement
- "KP offers proper abstraction, which disentangles motion perturbations and semantics changes"  How is it assessed in the experiments?
- The initial part of Sect. 3.1, where the amount of KP of each type is introduced, is a bit unclear to me. How such numbers are derived?
- “...we limit the criteria of KP as the indicator signs to minimize the need for human-defined standards (e.g., numerical criteria on the closeness of two joints) for objectivity and actuality “ I can not understand this statement
- When describing the way the different types of KP are extracted, it is often reported: “After filtering...”. However, I missed what this filtering is
- I am not sure I understand the difference between PRPP and LOP, the formula is apparently the same
- Sec. 4.1: are these preliminaries reporting things already introduced in the previous sections? How are they related to them?
- The description in “Model structure” in Sect. 4.2 is a bit dense and technical, I’m not sure it favours the understanding of a reader not fully familiar with the tools
- Self-supervision: I did not get how the self-supervised approach is designed (in particular, what is the task to be addressed for the self-supervision)
- "Moreover, most current motion generation evaluations are performed on datasets (Guo et al., 2022a; Plappert et al., 2016; Ji et al., 2018) with considerable complex everyday actions, further increasing the difficulty. " This should be better justified: in what sense the proposed approach is an advancement? And how it would extend to more complex actions?
- I fail to understand how the accuracy is computed. Giving an intuition, maybe with an example, would be beneficial
- “For each prompt, we generate one sample considering the annotation cost. We claim that the models should generate natural text-matching motion most of the time so that the one-sample setting would not hurt the fidelity of our user study.” I might misunderstand the statement, but I don’t think just one sample is enough to make considerations on the general behaviour of the method
- In Tab. 5 the performance w/o Body KS seems slightly better on the accuracy. It would be interesting to provide a comment on that

With their answer, the authors addressed my main concerns and clarified my doubts, I am willing to increase the rating

---

> ### Author Response · Authors · 2023-11-20
>
> Dear Reviewer 87dC:
>
> First and foremost, we extend our deepest gratitude for your thorough review and helpful advice in improving the paper. We sincerely appreciate for recognizing our work as fairly motivated. Herein, we respond to each of your concerns.
>
> 1. **Presentation.** Thanks for your kind advice on making the paper more understandable. For the metrics in Tab. 3, we explain the details in Appendix F.2 due to the page limit.
>
> 2. **Comparison to SOTA.**
> Notably, our text-to-motion model could be identified as improving the latent-based diffusion proposed in MLD with the newly proposed KP.
> This choice is made to conduct a fast and light-weight effectiveness evaluation of KP.
> Therefore, a proper comparison target should be MLD, upon which considerable advantages are shown in the user-study results and KPG.
> For T2M-GPT and MotionGPT, which are the current SOTAs, we would like to emphasize the different paradigms it adopted, as well as the model scale.
> Both of them tokenized motions into a codebook, coupled the codebook with text tokens, and then adopted a different pipeline as an auto-regressive decoder-only generative model, which resembles GPT.
> Also, they possess a considerably bigger model scale. For example, the 18-layer transformer is adopted by T2M-GPT for the text-to-motion task, however, in comparison, only a 4-layer denoiser and a 4-layer with a frozen CLIP text encoder are adopted in our method.
> Therefore, we would claim that despite the diverse paradigm and model scale, by incorporating KP into an MLD-like pipeline, we manage to achieve comparable performance with SOTAs, proving the efficacy of KP.
> Furthermore, given the motion quantization ability of KP as mentioned by Reviewer DPfm, also its effortless conversion to natural language, we are optimistic about incorporating it with T2M-GPT for future work.
>
> 3. **Clarifying the main target.** Thanks for your kind advice, and we will update our presentation in the revision.
>
> 4. **Visualization of more diverse motions beyond walking.** Please refer to the general response.
>
> 5. **Figures that are not explanatory enough.** Thanks for your kind advice, and we will polish the figure captions in the revision.
>
> 6. **"For objectivity and actuality, KP captures sign changes with minimal pre-defined standards."**
> By objectivity and actuality, we mean KPs are defined solely on the sign changes of the indicators calculated raw motion only, with little further human-defined thresholds, which could introduce unexpected subjectivity.
>
> 7. **Proper abstraction of KP.** In the first place, the proper abstraction is a design motivation of KP. Compared to raw motion, it is designed as more abstract with interpretable atomic Phrases. Compared to natural language, it tends to preserve more detailed information though not as concise as text descriptions, which could be shown by the higher diversity while keeping competitive consistency as shown in the user studies.
>
> 8. **The derivation of the number of KPs.** Here we list the details for KP filtering. These will be updated in the appendix.
>     - Position Phrase (PP). It is composed of 14 joints (except the pelvis and both eyes) and 3 reference vectors with originally 42 Phrases. Since there are joints that define the reference vectors (e.g. hips for the leftward vector). Specifically, shoulders and hips are excluded for the leftward vector, and hips are excluded for the upward vector, resulting in 42-6=36 PPs.
>     - Pairwise Relative Position Phrase (PRPP). It is composed of 136 joint pairs and 3 reference vectors with originally 408 Phrases. 24 joint pairs that are linked by limbs are filtered out. 30 Eye-related pairs are filtered out except (the left eye, and right eye) because the others are covered by head-related pairs. 4 triplets that the relationship barely changes along the reference vector (e.g., right knee, left hip, and leftward vector) are filtered out. These result in 408-24*3-30*3-4=242 PRPPs.
>     - Pairwise Distance Phrase (PDP). It is composed of 105 joint pairs (except for both eyes). 24 pairs that are linked by the human body are filtered out, resulting in 81 PDPs.
>     - Limb Angle Phrase (LAP). The arms, legs, and their link with the upper body are included as 8 LAPs.
>     - Limb Orientation Phrase (LOP). The 8 arm and leg limbs, head, collarbones, hips, torsos, and upper body, paired with the 3 reference vectors, result in 45 LOPs in total. 12 pairs that barely change along the reference vector (e.g., left-right hip, leftward vector) are filtered out, resulting in 45-12=33 LOPs.

---

> > ### Author Response · Authors · 2023-11-20
> >
> > 9. **"...we limit the criteria of KP as the indicator signs to minimize the need for human-defined standards (e.g., numerical criteria on the closeness of two joints) for objectivity and actuality."**
> > The inherent idea is that for different motions, different joints, and different people, the definition of proximity could be varied. How close is close for two joints?
> > If 15 centimeters is set as a criterion, it might be okay for feet, but for hands, it could be too large.
> > Given this, we adopt indicator signs as the only criteria.
> >
> > 10. **KP filtering.** Please refer to QA-8.
> >
> > 11. **PRPP and LOP.** Sorry for the ambiguity. For PRPP, the joint pairs that are not linked by the skeleton are considered. While for LOP, we focus on the human limb orientations. We will update this in the revision.
> >
> > 12. **Section 4.1.** It introduced the representations for the motion understanding system based on motion and KP, which is first mentioned in Sec. 1. The KP has been introduced in Sec. 3.
> >
> > 13. **Readability of Section 4.2.** Thanks for your kind advice. We would keep revising it to make it more readable.
> >
> > 14. **Self-supervision.** The KP-Motion joint space is trained with the task of reconstructing missing KP given corrupted KP sequences, which is a simple self-supervised objective as illustrated in Fig. 3.
> >
> > 15. **"Moreover, most current motion generation evaluations are performed on datasets (Guo et al., 2022a; Plappert et al., 2016; Ji et al., 2018) with considerable complex everyday actions, further increasing the difficulty."**
> > We argue that existing datasets with complex everyday actions are helpful, however, they are also hard to evaluate since there exists no reliable metrics on whether the generated motion is consistent with the given text prompt, especially when the prompt contains complex semantics. Instead, KPG provided a reliable metric for semantic consistency.
> > Though the current KPG contains mostly simple prompts, we have included temporal compositions of KP, while the concurrent composition of KP is also easy to achieve.
> > More complex compositions could explored for more complex actions, which we believe is worth future exploration.
> >
> > 16. **Accuracy computation.** Given the generated motion X with t frames and the target KP $c_i \in [-1, 0, 1]$, we first extract the KP sequence corresponding to $c_i$ as $C_i \in [-1,0,1]^t$. Then,  $c_i \in C_i$ is recognized when there is the sequence $c_i, c_i, c_i, c_i, c_i$ is a consecutive sub-sequence of $C_i$. For PRPP and LOP evaluation, the sequence $c_i^0, c_i^0, c_i^0, c_i^0, c_i^0, c_i^1, c_i^1, c_i^1, c_i^1, c_i^1$ is the expected sub-sequence. We will update the details in the PDF.
> >
> > 17. **"For each prompt, we generate one sample considering the annotation cost. We claim that the models should generate natural text-matching motion most of the time so that the one-sample setting would not hurt the fidelity of our user study." I might misunderstand the statement, but I don't think just one sample is enough to make considerations on the general behavior of the method.**
> > Here we clarify that the user study is designed with strict criteria with no tolerance for failure in semantic consistency. For general behavior modeling, we emphasize the considerably larger user-study scale (600 vs. less than 50 different prompts in previous works), which enables us to evaluate the models' behavior across a wide range of text prompts.
> >
> > 18. **Accuracy w/o Body KP.** Sorry for the typo. The result should be 52.04. Since there are only three Body KPs, removing them only results in a marginal performance decrease.

---

> > > ### Author Response · Authors · 2023-11-20
> > >
> > > Thank you again for your detailed and thorough feedback. We will incorporate the corresponding modifications and expansions in the revision. In addition, the corresponding code and model weights will be open-source to ensure replication. If there are any further questions, please inform us. We would be very happy to do anything we can that would be helpful in the time remaining. Thanks!

---

> > > ### Comment · Reviewer_87dC · 2023-11-22
> > >
> > > I sincerely thank the authors for their detailed comments on my concerns

---

### Author Response · Authors · 2023-11-20
**General Response**

Dear Area Chairs and Reviewers:

Thanks for your valuable reviews and insightful comments, which have helped us improve our paper. In the initial reviews, most reviewers found our work with **a good contribution** and **a good presentation**. We are glad that the proposed KP is identified as **well-thought-out** and **well-principled**.

In general response, we provided some new qualitative visualization samples as recommended by most reviewers at [**this link**](https://streamable.com/0onrex).
The video is also updated in the supplementary materials.

1. **Qualitative results with the standard metric values**. We further provide some new qualitative results to demonstrate some failure cases of the current standard metrics. R-precision could mistakenly assign relatively high metric values to a non-matching and unnatural motion. While it also sometimes fails to recognize matching motions. A potential reason is the pre-trained motion-text matcher overfitted to some specific patterns. For KPG, a similar phenomenon could be observed more often, which contains many out-of-distribution samples for the matcher. As shown, the standard R-Precision could fail to evaluate the authentic performance of motion-semantic consistency, while our model could perform better than R-Precision exhibits.

2. **Qualitative KPG failure mode analysis**. In Appendix F.3.3 and Figure 10, we include a typical user-studied KPG failure case of current models. where redundant motion is generated. To demonstrate more, we provide more failure cases in the attached video, from both accuracy and user studies. For accuracy, a major failure mode is due to the cases like "right shoulder is in front of left foot, then behind left foot". These are simple body joint relations like exercising instructions, however, not usually explicitly described in general datasets. This reveals that current models tend to be sub-optimal in real understanding of the human body structure. Also, previous efforts might even corrupt these. For user study, there are two major failure modes. First, our model could sometimes generate a motion with limited amplitude for target KPs, which could be captured by KP-base accuracy but humans could lose track of them. Second, previous models like T2M-GPT generate redundant motions for simple prompts, which could confuse users.

3. **More visualizations for interpolation and modification**. We provide some visualizations in the updated video. For interpolation, we adopt a more difficult setting than reported in the main text, where only the first frame and the final frame (red) are given. Our model could interpolate reasonable motion toward the target.

Detailed responses to individual concerns are included in the comments. We will keep on incorporating the corresponding modifications and expansions to the PDF in the revision.
Thank you again for your time and efforts paid to your helpful feedback. In addition, the corresponding code and model weights will be open-source to ensure replication. If there are any further questions, please inform us. We would be very happy to do anything we can that would be helpful in the time remaining. Thanks!

---

### Meta-Review · Area_Chair_pfUh · 2023-12-11

**Metareview:**

The paper considers the problem of abstracting low-level human motion to a semantic representation of motion, one that is challenging due to the disparate nature of the modalities. The paper proposes Kinematic Phrases (KP), an intermediate representation of motion as objective kinematic "facts" (features) related to the state and relationship between parts of the human body, which facilitates motion interpretability. The paper describes a VAE-based model that learns to align a latent motion space (via Motion VAE) with a latent KP space (via KP VAE), and shows that that the model enables motion interpolation, generation, and modification. The paper then proposes Kinematic Prompts Generation (KPG), a new benchmark that enables a quantifiable evaluation of the consistency of generated motions.

The paper was reviewed by four reviewers, each of whom participated in the author-reviewer discussion. The reviewers agree that the paper addresses an important problem---the gap that exists between raw human motion data and action semantics remains a challenge for human motion understanding---and the realization of an abstraction (and the associated mapping) that facilitates interpretability and generality is of interest to researchers working on human motion understanding. As Reviewer DPfm emphasizes, the proposed KPG benchmark is similarly beneficial as it provides a quantifiable metric by which the semantic consistency of generated human motion can be judged. The reviewers also appreciate the fact that the KP method is easy to replicate, which will benefit others who may want to use the representation for downstream tasks.

Meanwhile, some reviewers raised concerns regarding the experimental results. Among them, Reviewer DPfm finds that the paper misses the opportunity to analyze the limitations of current models on the proposed KPG benchmark, which would help to clarify which aspects of the problem the proposed method succeeds at. Such an analysis, would underscore the contributions of the KPG benchmark. As it stands, the reviewer believes that the initial set of analyses, notably the FID and diversity results, provide little insight into the performance of existing methods. During the author-reviewer discussion period, however, the authors presented an additional analysis of three existing methods (in addition to KP) on KPG, which shows that existing methods do well on some metrics (achieving 100% performance in some cases), but struggle on others. A more thorough quantitative and qualitative analysis of existing methods using the KPG benchmark, including these recent results, would strengthen the paper's contributions.

**Justification For Why Not Higher Score:**

I am torn on this one. As several reviewers comment, there is value in this paper both in terms of the method that it proposes for abstracting human motion data and the new benchmark that it proposes, which Reviewer DPfm suggests is actually the most valuable contribution. However, the extent to which the authors analyze existing methods (as well as the proposed method) on this dataset is rather limited. As Reviewer DPfm notes, this is a missed opportunity on the part of the authors. Including a thorough quantitative and qualitative analysis would noticeably strengthen the paper.

**Justification For Why Not Lower Score:**

N/A

---

### Decision · Program_Chairs · 2024-01-16

Reject